# Prenatal Exposition to Different Immunosuppressive Protocols Results in Vacuolar Degeneration of Hepatocytes

**DOI:** 10.3390/biology12050654

**Published:** 2023-04-26

**Authors:** Aleksandra Wilk, Dagmara Szypulska-Koziarska, Dorota Oszutowska-Mazurek, Alexander Baraniskin, Joanna Kabat-Koperska, Przemyslaw Mazurek, Barbara Wiszniewska

**Affiliations:** 1Department of Histology and Embryology, Pomeranian Medical University, 70-111 Szczecin, Poland; aleksandra.wilk@pum.edu.pl (A.W.);; 2Department of Hematology, Oncology and Palliative Care, Evangelisches Krankenhaus Hamm, 59063 Hamm, Germany; 3Department of Nephrology, Transplantology and Internal Diseases, Pomeranian Medical University, 70-111 Szczecin, Poland; 4Department of Signal Processing and Multimedia Engineering, West Pomeranian University of Technology in Szczecin, 71-126 Szczecin, Poland

**Keywords:** immunosuppressive drugs, vacuolar degeneration, liver, morphology, image analysis

## Abstract

**Simple Summary:**

Transplant recipients need to use immunosuppressive therapy in order to avoid organ rejection. These drugs, unfortunately, affect numerous organs, including liver. One of the commonly observed alteration in the hepatic tissue is vacuolar degeneration. Mostly, immunosuppressive therapy is based on three drugs from different groups: calcineurin inhibitors, mTOR inhibitors, IMDH inhibitors and glucocorticoids. There is scarse data regarding the influence of multidrug immunosuppressive therapy on condition of liver, including prenatal aspect. The aim of the current study was to compare the effects of prenatal exposition of different protocols of immunosuppressants on vacuolar degeneration in the hepatocytes of livers of rats. To our best knowledge, this is the first study concerning the effect of multidrug immnunosuppression administred in utero on hepatic tissue of rats’ offspring.

**Abstract:**

Immunosuppressive drugs are essential for transplant recipients, since they prolong proper function of graft; however, they affect the morphology and function of organs, including liver. One commonly observed alteration in hepatocytes is vacuolar degeneration. Numerous medications are contraindicated in pregnancy and breastfeeding, mostly due to a lack of data concerning their advert effects. The aim of the current study was to compare the effects of prenatal exposition to different protocols of immunosuppressants on vacuolar degeneration in the hepatocytes of livers of rats. Thirty-two livers of rats with usage of digital analysis of the images were examined. Area, perimeter, axis length, eccentricity and circularity regarding vacuolar degeneration were analysed. The most prominent vacuolar degeneration in hepatocytes in the aspects of presence, area and perimeter was observed in rats exposed to tacrolimus, mycophenolate mofetil and glucocorticoids, and cyclosporine A, everolimus with glucocorticoids.This is the first study that demonstrates the results of the influence of multidrug immnunosuppression distributed in utero on the hepatic tissue of offspring.

## 1. Introduction

In current transplantology there are four main groups of immunosuppressive drugs (IDs) which are used by solid-organ recipients [1,2,3,4,5]. These patients are often of a reproductive age and want to have biological children. These are (i) inhibitors of calcineurin (CNIs) containing cyclosporine A (CsA) and tacrolimus (TAC); (ii) inhibitors of mTOR consisting of sirolimus and everolimus (EVE); (iii) inhibitors of the inosine monophosphate dehydrogenase including mycophenolate mofetil (MMF), mycophenolate sodium (MMs) and azathioprine (AZA); and (iv) glucocorticoids, among the other prednisone (PRED) [1,2,3,4,5]. Regardless of numerous side effects that these drugs cause, they must be used on daily basis after transplantation (Tx), as they have antirejection properties [1,2,3,4,5].

Studies conducted on animals revealed that CsA had an advert effect on the reproductive system [6,7]. As far as the concentration of the metabolites of CsA was concerned, their high values were observed in the placenta [6], which indicated that this tissue contains enzyme-metabolizing CsA [7]. Classification of this ID performed by the FDA indicates that the risk for humans’ health cannot be ruled out, when CsA is used during pregnancy (Table 1). Another CNIs, tacrolimus, has the ability to cross the placenta as well. It is estimated that in-utero exposition to TAC is approximately 71% of maternal blood concentration [8]. A fetus has limited ability to metabolize TAC, which is most probably due to the fact that in a liver during the prenatal period, the most abundant cytochrome enzyme is CYP3A7. Concomitantly, it is known that CYP3A7 exhibits a high interindividual variability, whose molecular basis remains unknown [9]. However, the risk of major malformations is low for newborns from transplanted mothers. As far as the safety of TAC usage when breastfeeding is concerned, conclusions from studies as well as clinicians confirm that women on TAC can breastfeed, if monitoring of infant levels is available [10,11].

Mycophenolates are prodrugs that release mycophenolic acid (MPA), which acts as an inhibitor of inosine-5${}^{\prime }$-monophosphate dehydrogenase [20,21]. Mycophenolate acid has been categorized by the FDA as Category D., i.e., with positive evidence of human fetal risk [12] (Table 1).

Other IDs commonly used by clinicians are sirolimus and everolimus, both belonging to mTOR inhibitors [22,23]. When used per os, the mTOR inhibitors enter the cells and bind to FK binding protein12, which is their receptor. This complex inhibits mTOR, the downstream effector of a family of kinases originated by phosphatidylinositol 3-kinase which, together with a protein kinase B and through the mediation of mTOR, activates a cascade of kinases that originate numerous signaling pathways crucial for proliferation of T cells [24,25]. Still, there is scarse information regarding the safety of mTOR usage during pregnancy. According to the FDA, sirolimus and everolimus belong to the Category C (teratogenic risk cannot be ruled out because of lack of information) [12] (Table 1).

The last group of IDs, glucocorticoids, exert their effects through two main mechanisms of action: the classic genomic pathway and nongenomic pathway. Available glucocorticoids are effective anti-inflammatory agents which can influence the immune response, mostly the cellular, which is more susceptible in comparison to humoral immunity. Glucocorticoids are widely used for most autoimmune diseases, oncology, and transplantology [26]. Although these agents cross the placental barrier, 90% of the maternal dose of glucocorticoids is converted within the placenta into inactive products, while dexamethasone and betamethasone are less well metabolized. Therefore, the FDA classifies the fetal risk of glucocorticoids as the Category C, indicating that human risk cannot be ruled out [12] (Table 1).

The scheme presenting the mode of action of the abovementioned IDs is presented in Figure 1.

It is known that IDs affect the morphology and function of organs, including liver. One of the commonly observed alterations is vacuolar degeneration (VD) [27,28]. The hydropic alteration (vacuolar degeneration) results in a foamy cytosol of hepatocytes. It is a consequence of the oedema of the endoplasmic reticulum associated with an increase in intracellular water [27,28]. The vacuolization of the endoplasmic reticulum is associated with the expansion of the spaces of the reticulum, inside which there is an amorphous mass of low density towards electrons, and sometimes also components of bile. This may be accompanied with a decreased number or complete loss of ribosomes [27,28]. These lesions can be observed in cases of tissue hypoxia and toxic hapatopathies. The fatty change manifests itself in the form of pronounced vacuoles in the cytoplasm of hepatocytes, which is the result of the accumulation of lipids that can freely coalesce. These features of the appearance of the cytoplasm are often observed in lipidosis, but it was not observed in the current study [29,30]. It is often difficult to establish the underlying etiology of diffuse vacuolar hepatopathies. When hepatocytes are damaged, one of the responses is swelling and vacuolization. Hepatocellular vacuoles dilating the cytosol may contain fat, glycogen, intracellular water (oedema), or other metabolic products or intermediates. Vacuolar hepatopathies can occur in conjunction with hydropic degeneration, in which there is cytosolic oedema but a lack of pronounced vacuoles. A number of the reactive hepatopathies discussed above may be responsible for vacuolar hepatopathies or the lesions may arise from secondary chronic stress (presumably due to endogenous steroids) arising from the underlying diseases [29,30].

Numerous medications are contraindicated in the pregnancy and breastfeeding, mostly due to lack of data concerning their advert effects. The most numerous and prominent pathological alteration observed in current study was vacuolar degeneration. Therefore, the aim of this study was the comparison of the effects of prenatal exposition of different ID protocols on vacuolar degeneration in the hepatocytes of livers of rats litters.

## 2. Materials and Methods

Current study was based on material obtained from the Department of Nephrology, Transplantology and Internal Medicine from Pomeranian Medical University in Szczecin, Poland. This research was a continuation of a previous experiment on the effect of various immunosuppressive regimens on the morphology, apoptosis intensity and redox status in the native liver of pregnant Wistar rats [2,31]. Therefore, the presented design of current study was the same, as in aforementioned studies. First step of our analysis was morphological assessment of the livers from all 54 litters whose mothers were administered different regimens of IDs during the pregnancy. This analysis was performed with the usage of light microscopy. The next step was to analyse the hepatic tissue with usage of digital method. However, only 32 livers could be digitally analysed due to sufficient quality of image: 6 livers from the CONTROL group, 10 from TMG group, 9 from the CMG group, 7 livers from the CEG group. The doses of medicaments were based on the available literature [3]. Additionally, metabolic differences were also taken into consideration. The drug doses were based on data available in the literature [2,31] to reach a concentration within the therapeutic range. The study model is presented in Table 2. After delivery the treatment was stopped; therefore, no drugs were administrated during lactation (breastfeeding is not advised while taking immunosuppressive drugs).The consent of the bioethics committee and ethical standards were presented in [2,31].

### 2.1. Histology Examination of Vacuolar Degeneration

The liver samples were fixed in 4% paraformaldehyde and embedded in paraffin. Serial slices (3–4 μm thickness) of the liver of the control and experimental rats were mounted on poly-L-lysine coated glass slides and routinely stained with hematoxylin and eosin (H&E Staining Kit (HE; ab245880; Abcam, Cambridge, MA, USA). All histochemical reactions were carried out according to the manufacturer’s protocols. Next, the slides underwent general histological examination to investigate potential acute or chronic changes within the liver. The samples were independently examined by experienced histologists and pathomorphologists in blinded manner to avoid potential bias. Images from slides stained with HE were acquired with the usage of the microscope Axio Imager D1 (Carl Zeiss) and the AxioCam MRc5 camera with the 2572×1928 pixels resolution for HE under objective 40×.

### 2.2. Digital Analysis of Vacuolar Degeneration

Foamy appearance of cytoplasm of hepatocytes was described by pathologists as manifestation of VD. In order to assess the intensity of vacuolar degeneration, the segmentation was provided with the usage of Matlab.

Matlab with Image Processing Toolbox was used for image preprocessing. Image preprocessing (Figure 2) used RGB images and manually created masks were used to remove areas that did not contain relevant information, ex. areas of sinuses and central veins, located outside hepatocytes.

The first operation was contrast enhancement for standardisation of images (Figure 2). The second operation was image binarization using 50% threshold for Otsu’s algorithm. This binarization was independent process on all RGB channels. There were three binary planes for single images. The next operation was data fusion using logical operation that gave 1’s for all regions with white pixels of the input image. The last operation was removal of regions with areas <10 pixels. It was necessary since there was always some amount of noise in images and small object such as single pixels could not be processed by some shape descriptors (they are undefined or large errors could be achieved). Final results were based on binary image (segmented) that was processed using statistical methods or linear classifier.

### 2.3. Statistical Analysis

Ten predictors (Table 3) were selected for the shape analysis of white areas (corresponding with VD lesions in hepatocytes, visible as “holes” in microscopic HE images, that was caused partially by tissue processing before staining) in Figure 2, that were typically used for the image analysis purposes and corresponded to different features of object. Of note, there were some correlations between them and they were not orthogonal, so some redundancy was achieved. Optimally they are orthogonal; however, it was not possible to create such set for image data.

There were 5 predictors pairs with mean and standard deviation. Standard deviation was used as distribution dispersion estimator.

In order to compare the groups (CONTROL, TMG, CMG, CEG), merged data (all data of CONTROL, etc.) was used to check the possibility of using data modeling in statistical analysis. As it turned out, individual groups had different distributions; data analysis was also carried out.

### 2.4. Machine-Learning Approach

Linear classifier was selected due to small number of parameters, because more complex classifier could cause over-fitting. Overall classification dataset was very small (32 case and 4 classes), so only verification of classifier was possible. Testing classifier used testing and validation dataset, which is recommended if machine learning is not possible. It was limitation of current results, but could be valuable for evaluation of input images and possible important predictors.

The best results were achieved for highest accuracy and lowest number of predictors, as data set was small.

In this paper, metaoptimization of predictors was applied. There was $1023=2^{10}-1$ combination of predictors that could be created (‘−1’ corresponded to the cases with no predictors used, so this case was removed). All of them were tested for the calculation of accuracy. Every combination of parameters also delivered another important coefficient—number of predictors. Smaller values for the number of predictors was desired, since increasing number of them improved accuracy but reduced reliability of classifier due to very small data set. Due to this fact, it was not possible to select one, the most optimal classifier with particular predictors, because of two objective functions (accuracy related and number-of-parameters related).

We used our own automatic exhaustive search over all 1023 classifiers for analysis of the possibility determination differences between four classes CONTROL, TMG, CMG and CEG. Matlab with Statistics and Machine Learning Toolbox was used for processing data.

#### 2.4.1. Separate Classifiers

The first test concerned the examination of the possibility of classifying a specific class using individual classifiers. This means that for the CONTROL group, TMG, CMG and CEG, four undue linear classifiers could be proposed. Classifiers were taught by creating two classes (Table 4).

The purpose of this analysis was to determine the accuracy and the range of quantities and types of predictors that were required. Designation of a common group of predictors allowed determination of the most relevant data for the analysis.

#### 2.4.2. Common Predictors

This type of classifier used common set of predictors (shared between classifiers), so expected results were inferior to separate classifiers with their own individual sets of predictors. Instead of Accuracy, the Average accuracy from the four classifiers was used.

#### 2.4.3. Classifiers for Paired Groups

In this case, CONTROL and treatment groups were compared: TMG, CMG or CEG without merging them. It allowed to compare individual influence of TMG, CMG, CEG in relation to CONTROL (Table 5).

### 2.5. Vacuolar Degeneration

#### 2.5.1. Classifier for Vacuolar Degeneration

Single linear classifier was tested for the 1023 cases. All observations in CONTROL, TMG, CMG and CEG were merged together.

#### 2.5.2. Classifier for CONTROL and Merged TMG, CMG and CEG Group

Single linear classifier was tested for the 1023 cases. TMG, CMG anc CEG groups were merged. Only vacuolar degenerations were considered in this test.

## 3. Results

### 3.1. Distribution of Vacuolar Degeneration in All Examined Groups

Vacuolar degeneration was observed more frequently in treatment groups, in comparison to the control group (Figure 3 and Figure 4).

Additionally, the frequency of the presence of VD in hepatocytes between CONTROL group vs treatment groups in aspects of IDs protocols was compared. The results of this comparison were presented in Figure 3 and Figure 4.

### 3.2. Basic Global Estimators

In order to check the possibility of comparing global distributions, the Anderson–Darling test was used.

A two-sample Kolmogorov–Smirnov test for every pair CONTROL–TMG, CONTROL–CMG, and CONTROL–CEG indicated different distributions.

This approach was limited due to statistical differences between distributions. Due to this, a machine-learning approach was used for the deep analysis of data (Section 3.3 and Section 3.5.2). CONTROL and CEG groups were very scattered (very high values of standard deviations), but TMG and CMG were more then twice less scattered. It meant that smaller and larger lesions occured less frequently, since mean area for CONTROL, TMG and CEG were similar. Larger lesions occured in CEG and highest standard deviation corresponded to largest lesions. Perimeter parameters preserved similar relations. TMG, CMG and CEG were more circular comparing to CONTROL group, because Mean-circularity values were closer to one and the variability of this parameter was ten times lower comparing to CONTROL group. A similar trend was noticed for MAL and Eccentricity (Table 6).

### 3.3. Separate Classifiers

Two cases of CMG ($CMG_{1}$ and $CMG_{2}$) and CEG ($CEG_{1}$ and $CEG_{2}$) achieved with the same accuracy (Table 7).

This type of classifier was very demanding, as it assumed separation between particular group and other groups (alternative class).

There was no common favorite among predictors both in the entire set (Table 7) and in the groups of only mean and standard deviations (Table 8 and Table 9). Differentiation was based on many features of white lesions, although the analyzed features were related (to some extent) with each other; they were not orthogonal. In several cases, even for one class, CMG and CEG, there were alternatives to achieve identical accuracy results. It was possible to achieve Accuracy >87.50% using all 10 predictors (Table 7). Reduction to mean or standard deviations turned out to be problematic of the classification, because >81.25% (Table 8) or 71.88% (Table 9) Accuracy values were obtained.

### 3.4. Common Predictors

The machine-learning approach could be used for the analysis of important predictors (Table 10) and it was obtained using a common-predictors set for all classifiers. The histogram indicated that the Average accuracy was close to 90% and the worst was close to 75% (Figure 5). For common predictors, significant predictors for the highest Accuracy, of 88.28% (right tail of the histogram), were those that occured together in all cases (Table 10): Mean area and Mean circularity. The others form quite complex multidimensional relationships. These results corresponded to the previous statistical-analysis results.

However, when considering Accuracy values for single predictors (Table 11), the results were similar, indicating that a single predictor was not sufficient to describe image differentiation (white lesions), such as the Mean area and Mean circularity pair (Table 11—first line).

### 3.5. Classifiers for Paired Groups

For classification purposes, it was possible to achieve an Accuracy value of 100% (Table 12). There were three independent classifiers for paired groups: CONTROL-TMG, CONTROL-CMG and CONTROL-CEG. In this case, the lowest number of predictors was three. The result was linearly separable.

The most interesting were (CONTROL–TMG 3), (CONTROL–CMG 1) and (CONTROL–CEG 2) since they shared the Mean-perimeter predictor. Regrettably, the remaining pairs required further predictors to meet the case coverage requirement: Std.dev Perimeter, Std.dev. Area, Mean eccentricity, and Std. dev. eccentricity. This created a five-dimensional (5D) space which is difficult to assess but resulted in an Accuracy of 100%.

It was possible to reduce dimensionality at the expense of Accuracy when using common predictors for pairs, because the optimal criterion of 100% was a theoretical assumption, as it was possible that in some cases there was no drug effect due to individual characteristics. By reducing to three predictors and using the criteria of Accuracy overall > 90% and for CONTROL–TMG > 90%, CONTROL–CMG > 90% and CONTROL–CEG > 90% set of predictors could be obtained. Mean MAL, Std.dev. MAL and Mean eccentricity for an average Accuracy of 93.09%. The advantage was that it was easy to show linear separability in 3D space (Figure 6), but in the case of biological objects, although linear separability was visible, some cases may have been classified incorrectly.

A different set of predictors (Table 12) suggested that changes due to drugs could be different between CONTROL–TMG, CONTROL–CMG and CONTROL–CEG. It meant that geometrical changes in holes could be different. CONTROL–TMG proposed standard-deviation eccentricity as important factors, the next were the mean-area and standard-deviation perimeters. Two cases were interesting (No. 3 and 6): mean area and Std.dev. of area together with Std.dev. eccentricity; and mean perimeter and Std.dev. deviation of perimeter together with Std.dev. eccentricity; these two estimators were related to the single parameter (area and perimeter, respectively). Similar case was for CONTROL-CMG (perimeter estimators); however, the third one was mean eccentricity.

#### 3.5.1. Best Classifiers for Vacuolar Degeneration (Merged CONTROL, TMC, CMG and CEG)

Three cases of the highest accuracy and minimal number of predictors were shown in Table 13.

Results of separation for best predictors are shown in Figure 7.

Linear separation between images with and without vacuolar degeneration was obtained, although some cases may have been classified incorrectly due to the reasons described for Figure 6. Achieved accuracy was 96.88% for Mean MAL and Std.dev Circularity (Table 13 row 1). Similar results were achieved for pairs: (Mean perimeter, Mean eccentricity) and (Mean MAL, Std.dev. Area). In all the mentioned cases, one parameter was related to Mean and the second one to Std.dev. This could be explained by the larger widths of distributions of vaculoar degeneration (Figure 7–values).

#### 3.5.2. Classifier for Vacuolar Degeneration Using CONTROL and Merged TMG, CMG and CEG Group

There were cases (Table 14) that achieved 100% accuracy, but required five predictors. There were only twenty-four cases and this classifier was rather weak, so more examples were required for reliable analysis.

## 4. Discussion

Immunosuppression affects the morphology of blood vessels [5,32] and numerous organs, among them intestines [33], kidney [34] and liver [1,2,31,35]. Since, in current study, vacuolar degeneration was the most prominent and the most frequently observed alteration within hepatic tissue, we focused on the above-mentioned change. Vacuolar degeneration manifested as a foamy appearance of the cytoplasm of hepatocytes, being a result of endoplasmic reticulum oedema associated with an increase in intracellular water [36]. The current study was based on the comparison of the effects of prenatal usage of multidrug ID regimens in the aspect of VD in hepatocytes. To our knowledge, this study is unique due to the fact that it describes:(i)Multidrug immnunosuppression treatment;(ii)Effect of in-utero administration of IDs on the litters;(iii)VD as a single manifestation of liver injury.

Comparing the percentage number of rats from the perspective of presence vs absence of VD from each treatment groups, the regimens can be classified in descending order: TMG>CMG>CEG, 90%, 75%, 66%, respectively. Current results can be explained by the fact that according to the FDA, CsA, TAC, EVE and G belong to C group, which indicates that human risk cannot be ruled out. Mycophenolate mofetil belongs to D group, described as confirmed risk for human fetus. According to the literature, comparing the two used CNIs, TAC is more toxic than CsA [37]. Regarding the comparison between MMF and EVE, they exhibit a similar strength of action [38]. Although excessive vacuolar degeneration is considered a pathological alteration, in physiological state it is also present, however, at a lower percentage [36]. In the current study, vacuolar degeneration was noticed in 50% of cases in the control group.

Regarding the presence of VD, its distribution in hepatic tissue was noted as follows: the lowest frequency was found in the control, whereas the highest was observed in the TMG group. It seems, therefore, that the tacrolimus-based protocol induced VD at the highest frequency, in comparison to protocols based on other calcineurin ihibitor which is CsA. This trend persisted in the further analysis, where area, perimeter, circularity of VD in hepatocytes were compared. Obtained results suggest that IDs cause excessive presence of VD in hepatocytes. Moreover, current results correlate with our previous studies, where pregnant dams treated with TMG regimen were characterized by more pronounced pathomorphological changes than dams treated with protocols based on CsA [31]. Studies based on adult male rats showed that a TMG protocol affected mainly blood vessels [1]. This clearly indicates that an ID protocol based on tacrolimus seems to be more harmful in aspect of VD, in comparison to a regimen based on CsA. According to Ong et al. [37], tacrolimus is considered to be almost 100 times stronger than CsA. On the other hand, both the immature fetal liver and placenta can metabolise drugs. The lipophilic nature of CNIs allows them to passively diffuse across the placenta and enter the fetal circulation [39], as noted in one study by Flechner et al. [40]. In most reports, tacrolimus hepatotoxicity has been characterized by elevated levels of hepatocellular enzymes, either alone or with minimal cholestasis and hyperbilirubinemia. Furthermore, tacrolimus-induced cholestatic syndrome followed pediatric liver transplantation [41]. Cholestatis was also caused by CsA; however, a lowered dosage of the CsA inhibited pathological alterations within liver [42]. According to Lewis et al. [43], the blood concentration of CsA in the umbilical cord constituted 62% of the mother’s blood at the time of delivery, which can explain why the hepatic tissue of pregnant mothers had more severe alterations in comparison to their litters. Fatima et al. [44] observed histopathological alterations in liver architecture in TAC-treated rats. They noted altered liver architecture manifested by dilation of hepatic sinusoids, dearranged hepatocyte with cytoplasmic swelling and presence of glycogen globules [44]. According to available data, conversion from CsA to TAC is used in case of graft-rejection risk [45,46,47,48].

Interestingly, current comparison of the effects of the usage of multidrug regimens based on CsA showed that hepatocytes of rats treated with CMG protocol indicated the least number of rats with VD. Concomitantly, the area and the perimeter of VD in the CMG group were smaller in comparison to analogical parameters in rats receiveing CEG scheme. It should be emphasized that these two protocols differed in single co-drug only: everolimus (EVE)/mycophenolate mofetil (MMF). MMF exhibited a milding effect when combined with CsA, due to its antioxidant properties; therefore, it prolongs the viability of graft [49]. However, according to the FDA, MMF is contraindicated during the pregnancy [12,50,51].

Of note is the comparison between the CMG and CEG schemes, which showed more numerous and more prominent VDs in the hepatocytes of rats exposed to the protocol including EVE, which is an mTOR inhibitor. Regarding other pathological alterations, according to Chang et al. [52], EVE usage in mice resulted in decreased weight of the liver with a lowered accumulation of fat. There is data indicating that mTOR inhibitors increase the mortality of a fetus in animals. Nonetheless, no teratogenic effects were noticed either in rabbits nor rats. Still, there is scarse information regarding the safety of mTOR usage during the pregnancy. According to the FDA, EVE belongs to Category C (teratogenic risk cannot be ruled out). Available data indicates that EVE in multidrug immnunosuppression strongly disturbs the architecture and function of liver [31]. Regarding MMF, it improves both graft and patient survival. The properties of its active metabolite, mycophenolic acid, are diverse and include inhibition of de-novo purine synthesis and selective lymphocyte inhibition. It exhibits anti-tumoral, antiviral, anti-angioneoplastic, and vasculoprotective properties [38]. In this aspect of transplantation, the usage of a low-dose CNIs regimen with adjunctive MMF starting on the day of transplantation, the so-called ’de-novo MMF protocol’, might be a better alternative for organ recipients. According to the study of Boudjema et al. [53], the combination of low-dose TAC and MMF improves post-transplant renal function without increasing the risk of acute rejection [38,54,55].

Obtained data suggests that IDs affect the hepatic morphology of litters of mothers exposed to IDs during the pregnancy. It turned out that IDs exhibited hepatotoxic properties even when they are indirectly distributed, via the placenta. Although some ID combinations used during pregnancy lead to numerous pathological changes in the hepatic tissue, these may not be reflected in their litters.

Data analysis was also obtained using machine learning. Larger lesions in hepatocytes occurred in CEG, where perimeter parameters showed a similar pattern. What is more, it revealed that all treatment groups—TMG, CMG and CEG—contained more circular lesions comparing to the control group. This corresponded with the fact that vacuolar degeneration was observed more frequently in treatment groups. A similar pattern was observed for MAL and Eccentricity parameters. This revealed that a single predictor was insufficient for the description of images. The usage of drugs could cause changes in the image that could be quantified using morphological indicators of the image. It was observed that geometrical changes in white lesions due to drugs may differ between CONTROL and TMG, CONTROL and CMG and CONTROL and CEG. Linear separability was visible, but some cases were classified incorrectly because vacuolar degeneration is not only a pathological change and may occur also as functional change. The drug-resistance issue can be also taken into consideration, which may explain why there were a few cases without observed lesions in treatment groups.

## 5. Conclusions

Based on the results, it can be concluded that two protocols of IDs: TMG and CEG affected the morphology of hepatocytes the most with respect to VD parameters. Of note is that this is the first study that demonstrates the results of the influence of multi-drug immnunosuppression distributed in utero on hepatic tissue of offspring. These data may be an important indication for both patients and clinicians. Although most commonly used IDs are classified by the FDA as Category C—the risk to the fetus cannot be ruled out, benefits of use may exceed the risks.

## Figures and Tables

**Figure 1 biology-12-00654-f001:**
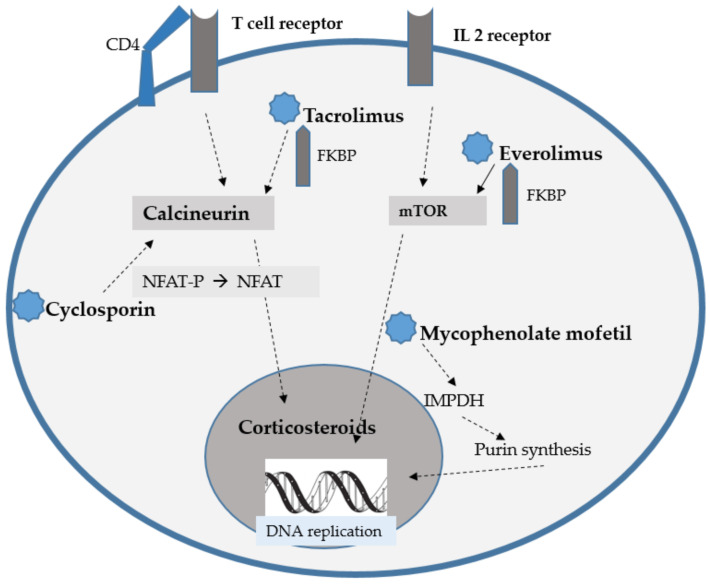
Mode of action of particular IDs, based on [1,2,3,4].

**Figure 2 biology-12-00654-f002:**
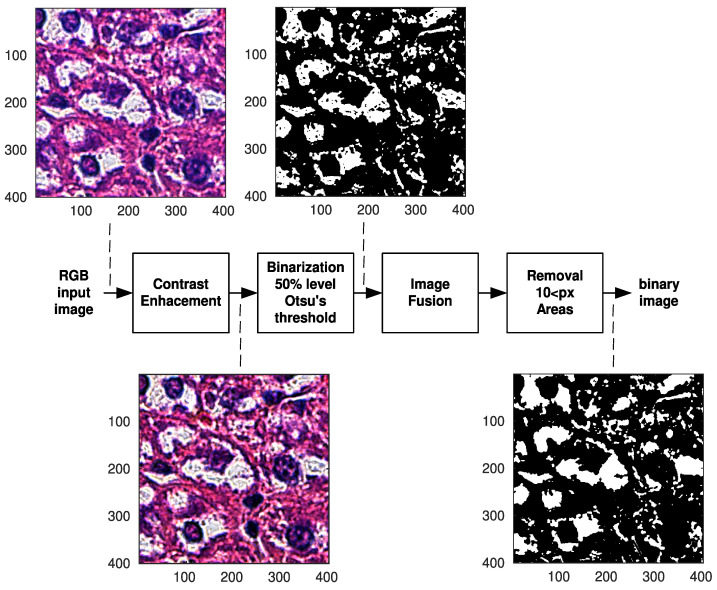
Preprocessing of images example.

**Figure 3 biology-12-00654-f003:**
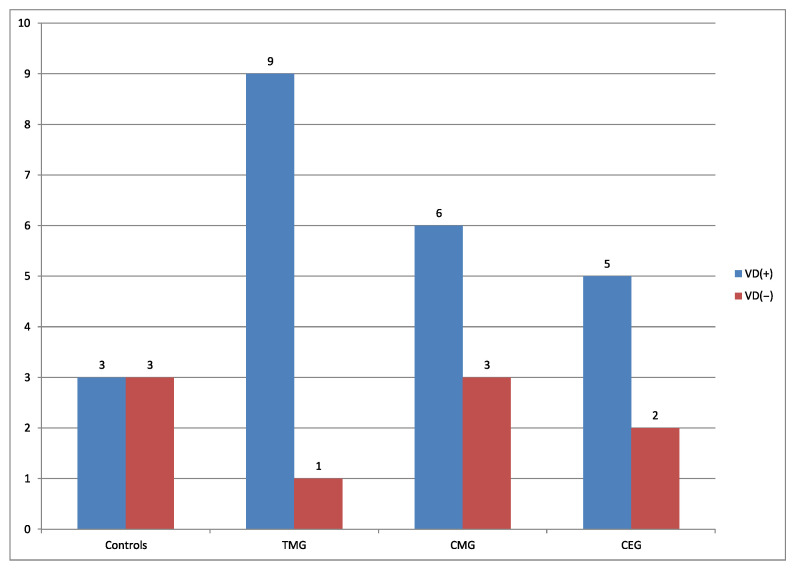
Number of cases of VD in hepatocytes of rats livers in all groups.

**Figure 4 biology-12-00654-f004:**
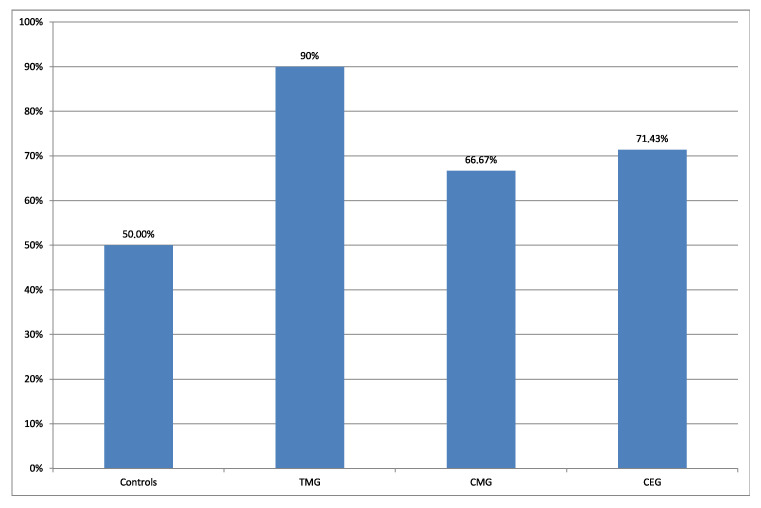
Percentage of VD in hepatocytes of rats livers in all groups.

**Figure 5 biology-12-00654-f005:**
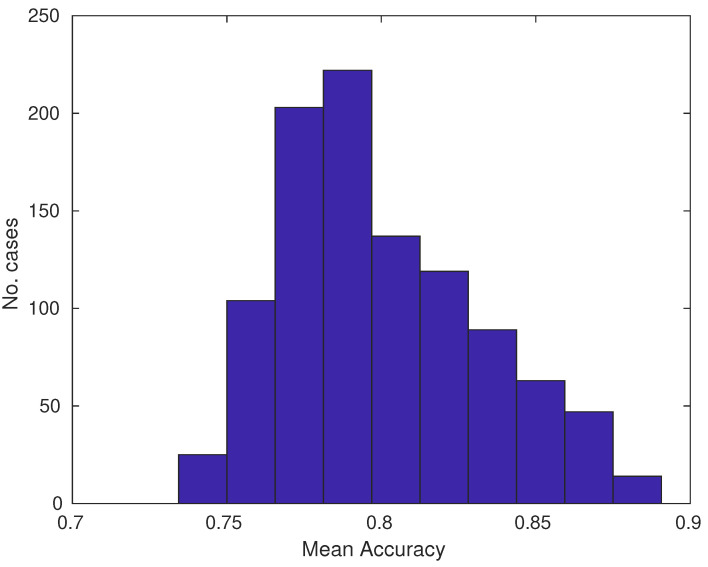
Histogram of mean accuracy for common predictors.

**Figure 6 biology-12-00654-f006:**
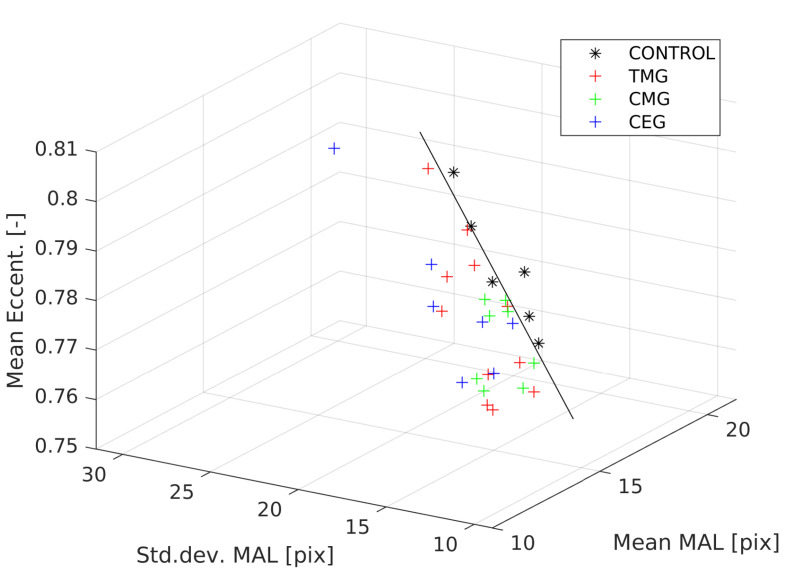
Linear separation for Mean MAL, Std.dev. MAL and Mean eccentricity predictors between CONTROL and merged group: TMG, CMG and CEG.

**Figure 7 biology-12-00654-f007:**
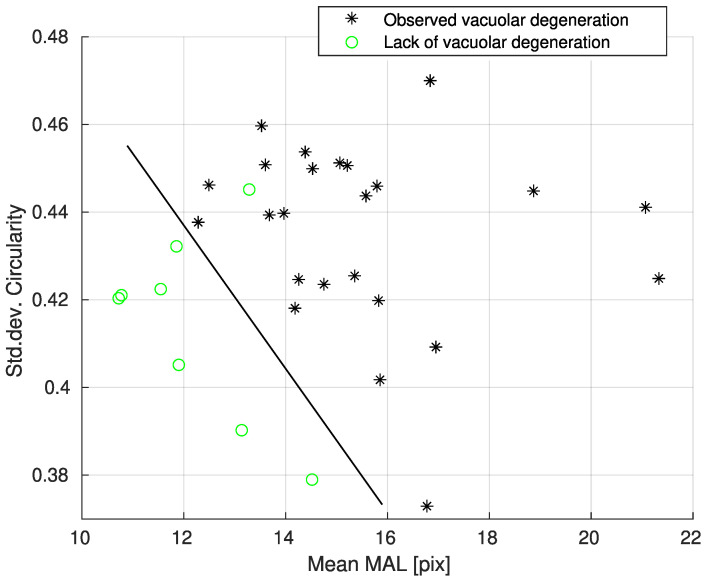
Linear separation for Mean MAL, Std dev. Circularity predictors between cases with present vacuolar degeneration and cases without lesions.

**Table 1 biology-12-00654-t001:** The drug administration classification according to teratogenic properties, based on [12].

DRUG	GROUP (According to FDA)	Effect On Fetus
Cyclosporine A	C/ Human risk cannot be ruled out	Increased mortality of animals in pre- and postnatal period [6,7] Decreased fetal weight with retardation of skeletal growth, cesarean delivery and hypertensive disorders [6,7] No side effects in the exposed children [13,14]
Tacrolimus	C/ Human risk cannot be ruled out	Increased risk of low birth weight and preterm birth [15,16]
mTOR inhibitors	C/ Human risk cannot be ruled out	Increased mortality of fetus in animals [12] No teratogenic effects have been noticed either in rabbits nor rats
Glucocorticoids	C/ Human risk cannot be ruled out	Hypertension and preeclampsia (at high doses during pregnancy) [12]
Mycophenolate	D/ Evidence of risk to human fetus	Increased risk of pregnancy loss and congenital malformations: external ear and other facial abnormalities including cleft lip and palate, and anomalies of heart, the distal limbs, kidney, esophagus [17,18,19]

**Table 2 biology-12-00654-t002:** The scheme of the experiment, based on [31].

CONTROL	CMG	TMG	CEG
(*n* = 24; 14 Males, 10 Females)	(*n* = 11; 6 Males, 5 Females)	(*n* = 12; 6 Males, 6 Females)	(*n* = 7; 3 Males, 4 Females)
	CsA 5 mg/kg bw/day	Tac 4 mg/kg bw/day	CsA 5 mg/kg bw/day
No drugs	Mmf 20 mg/kg bw/day	MMF 20 mg/kg bw/day	Eve 0.5 mg/kg bw/day
	G 4 mg/kg bw/day	G 4 mg/kg bw/day	G 4 mg/kg bw/day

**Table 3 biology-12-00654-t003:** Predictors used in classification process.

No.	Predictor
1	Mean area
2	Standard deviation of area
3	Mean perimeter
4	Standard deviation of perimeter
5	Mean major axis length (Mean MAL)
6	Standard deviation of major axis length
7	Mean eccentricity
8	Standard deviation of eccentricity
9	Mean circularity
10	Standard deviation of circularity

**Table 4 biology-12-00654-t004:** Data merging.

Class 1	Class 2 (Alternative, with Merged Cases)
CONTROL	TMG + CMG + CEG
TMG	CONTROL + CMG + CEG
CMG	CONTROL + TMG + CEG
CEG	CONTROL + TMG + CMG

**Table 5 biology-12-00654-t005:** Predictors used in classification process.

Class 1	Class 2 (Alternative)
CONTROL	TMG
CONTROL	CMG
CONTROL	CEG

**Table 6 biology-12-00654-t006:** Merged estimators.

	Mean	Std.dev.	Mean	Std.dev.	Mean	Std.dev.	Mean	Std.dev.	Mean	Std.dev.
	Area	Area	Perimeter	Perimeter	MAL	MAL	Eccent.	Eccent.	Circul.	Circul.
CONTROL	88.4908	231.7392	36.9539	53.0056	14.3227	14.3003	0.7883	0.1525	0.9536	0.4238
TMG	99.1502	91.9057	39.1379	18.0835	14.7395	2.7731	0.7769	0.0056	0.9871	0.0274
CMG	77.3647	73.3464	35.2019	16.4659	13.6196	2.3663	0.7762	0.0065	0.9927	0.0233
CEG	140.1070	402.5104	45.0806	69.0216	15.9124	7.9298	0.7751	0.0063	0.9770	0.0103

**Table 7 biology-12-00654-t007:** Best predictor combinations due to accuracy (‘x’-used estimator).

Class 1	Mean	Std.dev.	Mean	Std.dev.	Mean	Std.dev.	Mean	Std.dev.	Mean	Std.dev.	Accuracy
	Area	Area	Perimeter	Perimeter	MAL	MAL	Eccent.	Eccent.	Circul.	Circul.	
CONTROL		x			x	x	x				96.88%
TMG	x		x		x	x	x	x	x		87.50%
$CMG_{1}$	x	x	x	x		x	x		x	x	93.75%
$CMG_{2}$	x	x	x	x	x		x		x	x	93.75%
$CEG_{1}$		x		x							90.62%
$CEG_{2}$	x				x						90.62%

**Table 8 biology-12-00654-t008:** Best mean predictors combinations due to accuracy (‘x’-used estimator; ‘-’-not applicable).

Class 1	Mean	Std.dev.	Mean	Std.dev.	Mean	Std.dev.	Mean	Std.dev.	Mean	Std.dev.	Accuracy
	Area	Area	Perimeter	Perimeter	MAL	MAL	Eccent.	Eccent.	Circul.	Circul.	
CONTROL		-	x	-		-		-	x	-	87.50%
TMG	x	-	x	-	x	-	x	-	x	-	84.28%
CMG	x	-	x	-	x	-	x	-	x	-	81.25%
CEG	x	-		-	x	-		-		-	90.52%

**Table 9 biology-12-00654-t009:** Best standard deviation predictors combinations due to accuracy (‘x’-used estimator; ‘-’-not applicable).

Class 1	Mean	Std.dev.	Mean	Std.dev.	Mean	Std.dev.	Mean	Std.dev.	Mean	Std.dev.	Accuracy
	Area	Area	Perimeter	Perimeter	MAL	MAL	Eccent.	Eccent.	Circul.	Circul.	
CONTROL	-		-		-		-	x	-	x	87.50%
TMG	-	x	-	x	-	x	-		-		71.88%
$CMG_{1}$	-		-		-	x	-	x	-		81.25%
$CMG_{2}$	-	x	-		-		-	x	-		81.25%
CEG	-	x	-	x	-		-		-		90.62%

**Table 10 biology-12-00654-t010:** Best common predictors combinations due to accuracy (Accuracy = 88.28%) (‘x’-used estimator).

No.	Mean	Std.dev.	Mean	Std.dev.	Mean	Std.dev.	Mean	Std.dev.	Mean	Std.dev.	Mean
	Area	Area	Perimeter	Perimeter	MAL	MAL	Eccent.	Eccent.	Circul.	Circul.	Accuracy
1	**x**		x	x		x	x		**x**	x	88.28%
2	**x**		x	x	x		x		**x**	x	88.28%
3	**x**	x		x	x		x		**x**	x	88.28%
4	**x**	x		x	x	x		x	**x**		88.28%
5	**x**	x		x	x	x	x		**x**	x	88.28%
6	**x**	x		x	x	x	x	x	**x**		88.28%
7	**x**	x	x		x		x		**x**	x	88.28%
8	**x**	x	x		x	x		x	**x**		88.28%
9	**x**	x	x		x	x	x		**x**		88.28%
10	**x**	x	x		x	x	x		**x**	x	88.28%
11	**x**	x	x	x		x	x		**x**	x	88.28%
12	**x**	x	x	x	x		x		**x**	x	88.28%
13	**x**	x	x	x	x	x		x	**x**		88.28%
14	**x**	x	x	x	x	x	x		**x**	x	88.28%

**Table 11 biology-12-00654-t011:** Selected predictors due to accuracy (’x’-used estimator; ’-’-not applicable).

No.	Mean	Std.dev.	Mean	Std.dev.	Mean	Std.dev.	Mean	Std.dev.	Mean	Std.dev.	Mean
	Area	Area	Perimeter	Perimeter	MAL	MAL	Eccent.	Eccent.	Circul.	Circul.	Accuracy
1	x	-	-	-	-	-	-	-	x	-	78.12%
2	x	-	-	-	-	-	-	-	-	-	75.78%
3	-	x	-	-	-	-	-	-	-	-	77.34%
4	-	-	x	-	-	-	-	-	-	-	76.58%
5	-	-	-	x	-	-	-	-	-	-	75.78%
6	-	-	-	-	x	-	-	-	-	-	76.56%
7	-	-	-	-	-	x	-	-	-	-	76.56%
8	-	-	-	-	-	-	x	-	-	-	75.78%
9	-	-	-	-	-	-	-	x	-	-	76.56%
10	-	-	-	-	-	-	-	-	x	-	74.22%
11	-	-	-	-	-	-	-	-	-	x	75.00%

**Table 12 biology-12-00654-t012:** Best predictors for paired groups (‘x’-used estimator).

No.	Mean	Std.dev.	Mean	Std.dev.	Mean	Std.dev.	Mean	Std.dev.	Mean	Std.dev.	Mean
	Area	Area	Perimeter	Perimeter	MAL	MAL	Eccent.	Eccent.	Circul.	Circul.	Accuracy
CONTROL-TMG:											
1					x	x		x			100%
2				x	x			x			100%
3			x	x				x			100%
4	x					x		x			100%
5	x			x				x			100%
6	x	x						x			100%
CONTROL-CMG:											
1			x	x			x				100%
CONTROL-CEG:											
1		x			x		x				100%
2		x	x				x				100%

**Table 13 biology-12-00654-t013:** Best predictors for vacuolar degeneration (merged CONTROL, TMC, CMG and CEG) (‘x’-used estimator).

No.	Mean	Std.dev.	Mean	Std.dev.	Mean	Std.dev.	Mean	Std.dev.	Mean	Std.dev.	Mean
	Area	Area	Perimeter	Perimeter	MAL	MAL	Eccent.	Eccent.	Circul.	Circul.	Accuracy
1					x					x	96.88%
2			x				x				93.75%
3		x			x						93.75%

**Table 14 biology-12-00654-t014:** Best predictors for vacuolar degeneration (CONTROL and merged TMG, CMG and CEG group) (‘x’-used estimator).

No.	Mean	Std.dev.	Mean	Std.dev.	Mean	Std.dev.	Mean	Std.dev.	Mean	Std.dev.	Mean
	Area	Area	Perimeter	Perimeter	MAL	MAL	Eccent.	Eccent.	Circul.	Circul.	Accuracy
1			x		x	x		x	x		100%
2		x	x	x	x				x		100%

## Data Availability

Paraffin embedded tissue blocks are unique material and only histological glass slides stained in HE are available in Department of Histology and Embryology, Pomeranian Medical University.

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
