# Peer review of "Prenatal Exposition to Different Immunosuppressive Protocols Results in Vacuolar Degeneration of Hepatocytes"

_biology, 2023, doi:10.3390/biology12050654_

Round 1

Reviewer 1 Report

Aleksandra Wilk and colleagues investigated whether the prenatal use of different immunosuppressive protocols result in vacuolar degeneration of hepatocytes. It was found that protocols of IDs :TMG and CEG effected the most on morphology of hepatocytes in aspect of presence, area and perimeter of VD. It is more relevant because it is the first study to investigate the effect of in utero distributed multi-drug immunosuppression on the liver tissue of the offspring. However there are some problems:

(1) The abstract needs to be more than just a description of the background and purpose,and it also needs to have methods and results;

(2) The results should be divided into subsections with subheadings;

(3) The interpretation of the graphs and tables in the results is too simple, e.g. "vesicular degeneration was more frequent in the treatment group compared to the control group" (Figure 4);

(4) How many in a litter and how many in each group is not clearly written;

(5) 24 in the normal control group in the method to only 5 in Figure 4, and the numbers were different before and after each group;

(6) Need to explain in the discussion which drug is likely to be the most toxic (only one drug in each group differs);

(7) Is there a reasonable explanation for the presence of 40% VD in the normal group?

(8) What was the intra- and inter-litter VD in each dosing group?

Author Response

------------------------------------------ Reviewer’s Comments --------------------------------

Dear Reviewer 1:

Thank you for your advice and constructive comments concerning our manuscript entitled, “Prenatal exposition to different immunosuppressive protocols results in vacuolar degeneration of hepatocytes”. We have carefully considered your suggestions, and we hope that our response meet with your approval. All the changes are marked in the text with red and green colours.

Reviewer 1:

Aleksandra Wilk and colleagues investigated whether the prenatal use of different immunosuppressive protocols result in vacuolar degeneration of hepatocytes. It was found that protocols of IDs :TMG and CEG effected the most on morphology of hepatocytes in aspect of presence, area and perimeter of VD. It is more relevant because it is the first study to investigate the effect of in utero distributed multi-drug immunosuppression on the liver tissue of the offspring. However there are some problems:

(1) The abstract needs to be more than just a description of the background and purpose,and it also needs to have methods and results;

We wish to thank Reviewer for his/her useful comments. The abstract has been changed, we have added all the information that we were asked for.

(2) The results should be divided into subsections with subheadings;

Thank you very much for this comment, the results have beed divided into subsections with subheadings, according to your suggestions.

(3) The interpretation of the graphs and tables in the results is too simple, e.g. "vesicular degeneration was more frequent in the treatment group compared to the control group" (Figure 4);

We would like to thank for this comment. We have reconstructed this whole section, to make it more optimal and accurate. We hope it meets your approval.

(4) How many in a litter and how many in each group is not clearly written;

We would like to thank for this comment. We have reconstructed Material and Methods, where we described the number of examined livers from litters of each group, to make it more clear. We hope it meets your approval.

(5) 24 in the normal control group in the method to only 5 in Figure 4, and the numbers were different before and after each group;

We are grateful for this comment. We have reconstructed Material and Methods, where we dascribed the number of examined livers from litters of each group, to make it more clear;
First step of our analysis was morphological assessment of the livers from all 54 litters whose mothers were administered different regimens of IDs during the pregnancy. This analysis was performed with the usage of light microscopy. The next step was to analyse hepatic tissue with usage of digital method. However, only 32 livers could be digitally analysed due to sufficient quality of image:  6 livers from the control group, 10 from TMG group, 9 from the CMG group, 7 livers from the CEG group, Figure 5)”. We hope it meets your approval.

(6) Need to explain in the discussion which drug is likely to be the most toxic (only one drug in each group differs);

Thank you for this comment. We have fixed this part of the discussion, to make it more clear and accesible. However, the attenion should be paid to the fact, that there are 2 differences between TMG&CEG groups, not only 1, as it is between CMG&TMG or CMG&CEG. Therefore, we completed the discussion with the comparison of toxicity between Tacrolimus vs Cyclosporin A and Mycophenolate mofetil vs Everolimus.

(7) Is there a reasonable explanation for the presence of 40% VD in the normal group?

We are grateful for this important comment. It is interesting, that even in livers of healthy animals, there is always some vacuolar degeneration under physiological condition. We have emphasized it in the Discusion and suported with the proper literature ; „Although excessive vacuolar degeneration is considered as pathological alteration, in physiological state it is also present, however, in lower percentage [36 ]”.

8) What was the intra- and inter-litter VD in each dosing group?

W have prepared new tables and figures (Results) and we have expanded the  Methods regarding digital analysis, to present the differences between each dosing group and within the each group regarding the VD. We hope, it meets you approval.

We wish to thank Reviewer for his/her useful comments.

We would like to thank the reviewer for helpful comments. We hope that this manuscript is acceptable for publication in Biology.

Author Response

Szczecin, 13rd April 2023

------------------------------------------ Reviewer’s Comments --------------------------------

Dear Reviewer 2:

Thank you for your advice and constructive comments concerning our manuscript entitled, “Prenatal exposition to different immunosuppressive protocols results in vacuolar degeneration of hepatocytes”. We have carefully considered your suggestions, and we hope that our response meet with your approval. All the changes are marked in the text with red and green colours.

Reviewer 2:

  1. Grammatical errors, typos, inappropriate usage of articles, and punctuations should be

thoroughly checked for in the entire manuscript.

We wish to thank Reviewer for his/her useful comments. The English correction has been done, including grammatical aspect.

  1. The title of the article should reflect the finding of the manuscript rather than being a question.

Thank you. The title of the manuscript has been changed.

  1. The abstract should be modified to include summary of the methods and results of the study.

We wish to thank Reviewer for his/her useful comments. The abstract has been changed, we have added all the information that we were asked for.

  1. Reformat Figure 1 into a table format and include column headers. Information from ID can be a separate column. Information described in lines 52-100 can be succinctly summarized in the table.

Thank you. We have changed Figure 1 into Table.

  1. Reformat Figure 3 into a table format and include column headers.

Thank you. We have changed Figure 3 into Table.

  1. In line 144, how were the doses calculated to be equivalent to those in human therapy?

We have added information in Methods section: “The doses of medicaments were based on the available literature [3]. Additionally, metabolic differences were also taken into consideration. The drug doses were based on data available in the literature [2,31 ] to reach a concentration within the therapeutic range. The study model is presented in (Table 2). After delivery the treatment was stopped, therefore no drugs were administrated during lactation (breastfeeding is not advised while taking immunosuppressive drugs)”.

  1. Explain why areas smaller than 100 pixels were selected as artefacts as mentioned in lines 170-171. Add images to support the statement. Why has this value changes to 1000 pixels as

mentioned in 188?

Thank you for that comment. According to your suggestion- this has been improved.

  1. How were the area and perimeter calculated from the image? Elaborate on the software and pipeline used in the methods section.

Thank you for that comment. According to your suggestion- this has been improved.

  1. Why was regression line used in preference to other mathematical models? Have the authors tested other fitting curves?

Thank you for that comment too. We have improved the mathematical models (Methods, Results), to make our results more reliable and clear. We hope it meets your approval.

  1. For Figure 4 and 5, add error bars and p-values. Label y-axis. 3D visualization does not add any value.

Thank you, we have improved it.

  1. For figure 4 and 5, how has the y-axis value been calculated?

Yes, it has been calculated.

  1. Results in each of the figures from 8-13 can be succinctly presented in one plot instead of four plots. The results should be explained quantitatively in the text and not qualitatively.

We wish to thank Reviewer for his/her useful comments. It has been changed.

  1. How many images were analyzed to obtain results in Table 1?

We have analysed 32 imagaes, according to out Methods: First step of our analysis was morphological assessment of the livers from all 54 litters whose mothers were administered different regimens of IDs during the pregnancy. This analysis was performed with the usage of light microscopy. The next step was to analyse hepatic tissue with usage of digital method. However, only 32 livers could be digitally analysed due to sufficient quality of image: 6 livers from the control group, 10 from TMG group, 9 from the CMG group, 7 livers from the CEG group, Figure 5)”.

  1. Why have the authors chosen only area and perimeter as the parameters for measuring VD? Have they tried using other morphological characteristics such as shape?

Thank you for this comment. We have added more parameters (Table 3).

We wish to thank Reviewer for his/her useful comments.

We would like to thank the reviewer for helpful comments. We hope that this manuscript is acceptable for publication in Biology.

Reviewer 3 Report

The authors examined and compared vacuolar degeneration of rat litters' hepatocytes post prenatal exposition of three different regimens with immunosuppressive drugs, and demonstrated that tacrolimus-based treatment protocol induced highest frequency of vacuolar degeneration. These observations are helpful for evaluating side effects of immunosuppressive regimens. This manuscript will benefit from improving the following parts:

1. For a research article, it usually provides little information when the title is presented as a question. In addition, the current Abstract is mainly composed of background and one sentence of study aim, but no major conclusions and indications are stated at all. It is recommended that the authors re-write the title and the Abstract in a more proper and informative way. 

2. On page 2 Lines 47-49, please add appropriate references for the statement "Studies conducted on animals... The scientists have observed among...".

3. Histology examination and evaluation are better carried out in a blinded manner to avoid potential bias. The authors should indicate in the method section whether they evaluated in a blinded or non-blinded manner.

4. Certain figures are recommended to be merged into one to be more concise and easy to cross-compare. For instance, Fig 4 and 5 showed the absolute number and the frequency of rat litters that had vacuolar degeneration, respectively. Since the two figures demonstrated a similar point, it is recommended to simply combine these two figures as one (i.e. Fig 4a, 4b). Similarly, combine Figs 6 and 7 into one, Figs 8-10 into one, and Figs 11-13 into one. The graph sizes in Figs 8-13 need be shrunk while font sizes are maintained to allow all four groups fit into one row, and there would be three rows (i.e. perimeter, area, area/perimeter relation) in one figure. 

Author Response

------------------------------------------ Reviewer’s Comments --------------------------------

Dear Reviewer 3:

Thank you for your advice and constructive comments concerning our manuscript entitled, “Prenatal exposition to different immunosuppressive protocols results in vacuolar degeneration of hepatocytes”. We have carefully considered your suggestions, and we hope that our response meet with your approval. All the changes are marked in the text with red and green colours.

Reviewer 3:

The authors examined and compared vacuolar degeneration of rat litters' hepatocytes post prenatal exposition of three different regimens with immunosuppressive drugs, and demonstrated that tacrolimus-based treatment protocol induced highest frequency of vacuolar degeneration. These observations are helpful for evaluating side effects of immunosuppressive regimens. This manuscript will benefit from improving the following parts:

  1. For a research article, it usually provides little information when the title is presented as a question. In addition, the current Abstract is mainly composed of background and one sentence of study aim, but no major conclusions and indications are stated at all. It is recommended that the authors re-write the title and the Abstract in a more proper and informative way. 

Thank you very much for this comments. We have changed the title, as well as the abstract.

  1. On page 2 Lines 47-49, please add appropriate references for the statement "Studies conducted on animals... The scientists have observed among...".

Thank you very much, we have added the references.

  1. Histology examination and evaluation are better carried out in a blinded manner to avoid potential bias. The authors should indicate in the method section whether they evaluated in a blinded or non-blinded manner.

Thank you very much for this comments . The samples were independently examined by experienced histologists and pathomorphologists in blinded manner to avoid potential bias.

  1. Certain figures are recommended to be merged into one to be more concise and easy to cross-compare. For instance, Fig 4 and 5 showed the absolute number and the frequency of rat litters that had vacuolar degeneration, respectively. Since the two figures demonstrated a similar point, it is recommended to simply combine these two figures as one (i.e. Fig 4a, 4b). Similarly, combine Figs 6 and 7 into one, Figs 8-10 into one, and Figs 11-13 into one. The graph sizes in Figs 8-13 need be shrunk while font sizes are maintained to allow all four groups fit into one row, and there would be three rows (i.e. perimeter, area, area/perimeter relation) in one figure. 

Thank you very much for this useful comment, we have greatly changed and improved the tables and figures, to make them more consius and understandable. The great part of Methods and Results have been changed and expanded. We hope it finds your approval.

We wish to thank Reviewer for his/her useful comments.

We would like to thank the reviewer for helpful comments. We hope that this manuscript is acceptable for publication in Biology.

Round 2

Reviewer 1 Report

The author answered my questions and added content. I think these are all reasonable. Some minor issues do not affect the overall quality of the article.

Author Response

1
------------------------------------------ Reviewer’s Comments --------------------------------
Reviewer 1:
The author answered my questions and added content. I think these are all reasonable. Some minor issues do not affect the overall quality of the article.
Thank you for review of our article: “Prenatal exposition to different immunosuppressive protocols results in vacuolar degeneration of hepatocytes”.
We hope that this manuscript is acceptable for publication in Biology.

Reviewer 2 Report

The authors have addressed a few comments mentioned earlier. However, overall the manuscript is very difficult to follow in it’s current format. The tables and the analysis performed are still unclear and the rationale behind using certain analysis procedures is hard to understand. Here are a few specific points that still need to be addressed:

1.       Line 1 of abstract and introduction are repetitive. Though the authors have added more information in the methods section, it is very hard to follow the procedure that was used for the analysis.

2.       As mentioned in the previous review, information described in lines 52-100 can be succinctly summarized in table 1.

3.       Combine Figure 2 and figure 3. Provide corresponding images for each step mentioned in Figure 2.

4.       Figure 3 needs scale bars.

5.       In line 152, how are holes defined?

6.       How ere the predictor pairs in table 3 selected?

7.       Can the authors show data to support the statement in line 170-171?

8.       How and why was class 2 created in table 4?

9.       Figure 4 and figure 5 need y-axis to be labelled. Add error bars and p-values as mentioned in previous review.

10.   The control data in Figure 4 and 5 is different from the previous version. How did the data change?

11.   Table 6-10 need to be described in detail. It is very hard to follow the tables with the current description.

12.   Results need to be thoroughly explained in the results section. They are mentioned in the discussion section.

13.   In figure 7, there is no visible separation between the groups. How do the authors explain the observation?

14.   Grammatical errors, typos, inappropriate usage of articles, and punctuations should be thoroughly checked for in the entire manuscript.

Author Response

1
RE: Manuscript ID
Biology 2273972
Szczecin, 19th April 2023
------------------------------------------ Reviewer’s Comments --------------------------------
Dear Reviewer 2:
Thank you for your advice and constructive comments concerning our manuscript entitled, “Prenatal exposition to different immunosuppressive protocols results in vacuolar degeneration of hepatocytes”. We have carefully considered your suggestions, and we hope that our response meet with your approval. All the changes are marked in the text with red and green colours.
The authors have addressed a few comments mentioned earlier. However, overall the manuscript is very difficult to follow in it’s current format. The tables and the analysis performed are still unclear and the rationale behind using certain analysis procedures is hard to understand. Here are a few specific points that still need to be addressed:
1. Line 1 of abstract and introduction are repetitive. Though the authors have added more information in the methods section, it is very hard to follow the procedure that was used for the analysis.
Thank you. The abstract has been improved.
2. As mentioned in the previous review, information described in lines 52-100 can be succinctly summarized in table 1.
Thank you. Table 1 has been changed and expanded according to your suggestions.
3. Combine Figure 2 and figure 3. Provide corresponding images for each step mentioned in Figure 2.
Thank you, it has been done.
4. Figure 3 needs scale bars.
Thank you for this comment. It has beed added.
5. In line 152, how are holes defined?
2
The
The wholewhole explanationexplanation hahass been been addedadded in in the the main main texttext of manuscriptof manuscript. . TThe he hholes corresponds oles corresponds to white lesion inside hepatocytes, described as vacuolar degeto white lesion inside hepatocytes, described as vacuolar degenneratioerationn (VD). They are visible as (VD). They are visible as white areas really similar to holes. The white white areas really similar to holes. The white areas do not contain materialareas do not contain material due to tissue due to tissue processing before HE staining.processing before HE staining.
6. How ere the predictor pairs in table 3 selected?
There are typical image features parameters used in computer vision for morphological
There are typical image features parameters used in computer vision for morphological analysis.analysis. They are available in Matlab IThey are available in Matlab Image Processing Toolbox. There are hundreds of mage Processing Toolbox. There are hundreds of morphological operators andmorphological operators and practically infinite number of combination of them, so we practically infinite number of combination of them, so we have have usedused basic operators and combine them in metaoptimbasic operators and combine them in metaoptimiizationzation to for testing complex relations to for testing complex relations (1023 cases).(1023 cases).
7. Can the authors show data to support the statement in line 170-171?
It is a
It is a bbasis asis of optimization with a multiple independent objective functions.of optimization with a multiple independent objective functions.
Adding additional parameters (objective function no.1) for fixed objective function no.2
Adding additional parameters (objective function no.1) for fixed objective function no.2 (responsible for the reduction(responsible for the reduction of classifier error) create redundant interpolator.of classifier error) create redundant interpolator.
There are books about Occam's razor, biased estimators and failure of using interpolation in
There are books about Occam's razor, biased estimators and failure of using interpolation in classifiers.classifiers. ShowingShowing ofof such data will adds a mess and are not informative.such data will adds a mess and are not informative.
8. How and why was class 2 created in table 4?
Class 2 is alternative to Class 1
Class 2 is alternative to Class 1
Mathematically:
Mathematically:
Class1=~Class2
Class1=~Class2
it means that classifier outputs logical results {0,1}
it means that classifier outputs logical results {0,1}
Class1=0 ; Class2=1
Class1=0 ; Class2=1
and
and
Class1=1 ; Class2=0.
Class1=1 ; Class2=0.
9. Figure 4 and figure 5 need y-axis to be labelled. Add error bars and p-values as mentioned in previous review.
Both figures shows counts.
Both figures shows counts.
Modified caption to:
Modified caption to:
"Number of cases of VD in hepatocytes of rats livers in all groups"
"Number of cases of VD in hepatocytes of rats livers in all groups" for clarification.for clarification.
Error bars and p
Error bars and p--values are not possible to add to this type of diagram. values are not possible to add to this type of diagram. This is the input, not This is the input, not the output of the analysis.the output of the analysis.
3
10. The control data in Figure 4 and 5 is different from the previous version. How did the data change?
Thank you for that comment. We did our best and we have examined more images from control group and therefore data has been changed.
11. Table 6-10 need to be described in detail. It is very hard to follow the tables with the current description.
Table shows particular combination of estimators.
Table shows particular combination of estimators.
Added legend: ’x’
Added legend: ’x’--used estimator; ’used estimator; ’--’’--not applicablenot applicable. . Discussion moved to results should Discussion moved to results should imprimprove readabilityove readability..
12. Results need to be thoroughly explained in the results section. They are mentioned in the discussion section.
T
Thank hank you fyou for this comment. It has been improved.or this comment. It has been improved.
13. In figure 7, there is no visible separation between the groups. How do the authors explain the observation?
Added line and clarified caption:
Added line and clarified caption:
"Linear separation for Mean MAL, Std.dev. MAL i Mean Eccentricity predictors between
"Linear separation for Mean MAL, Std.dev. MAL i Mean Eccentricity predictors between CONTROL and merged group: TMG, CMG and CEG"CONTROL and merged group: TMG, CMG and CEG"
14. Grammatical errors, typos, inappropriate usage of articles, and punctuations should be thoroughly checked for in the entire manuscript.
Thank you, it has been improved.
We would like to thank the Reviewer for helpful comments. We hope that this manuscript is acceptable for publication in Biology.

Round 3

Reviewer 2 Report

The authors have addressed most of the comments and only needs slight modifications to the language.